# Community Participatory Approach to Design, Test, and Implement Interventions That Reduce Risk of Bat-Borne Disease Spillover: A Case Study from Cambodia

**DOI:** 10.3390/tropicalmed11010007

**Published:** 2025-12-27

**Authors:** Dou Sok, Sreytouch Vong, Sophal Lorn, Chanthy Srey, Madeline Kenyon, Bruno M. Ghersi, Tristan L. Burgess, Marcia Griffiths, Disha Ali, Elaine M. Faustman, Elizabeth Gold, Jonathon D. Gass, Felicia B. Nutter, Janetrix Hellen Amuguni, Jennifer Peterson

**Affiliations:** 1STOP Spillover, Tetra Tech ARD Inc., Phnom Penh 12102, Cambodia; sreychanthy@gmail.com; 2JSI Research & Training Institute, Inc., Phnom Penh 12308, Cambodia; vongsreytouch@gmail.com; 3Provincial Department of Agriculture, Ministry of Agriculture, Phnom Penh 12000, Cambodia; lornsophal72@gmail.com; 4Department of Public Health and Community Medicine, Tufts University School of Medicine, Boston, MA 02111, USA; madelineannk@gmail.com (M.K.); jonathon.gass@tufts.edu (J.D.G.); 5Cummings School of Veterinary Medicine, Tufts University, Medford, MA 02155, USA; bruno.ghersi_chavez@tufts.edu (B.M.G.); tburgess@centerforwildlifestudies.org (T.L.B.); felicia.nutter@tufts.edu (F.B.N.); janetrix.amuguni@tufts.edu (J.H.A.); 6Center for Wildlife Studies, South Freeport, Maine, Camden, ME 04843, USA; 7Manoff Group, John Snow, Inc., Charlottesville, VA 22906, USA; mgriffiths_24@outlook.com; 8JSI Research & Training Institute, Inc., Arlington, VA 22209, USA; dishaali09@gmail.com (D.A.); alohalizgold@gmail.com (E.G.); 9School of Public Health, University of Washington, Seattle, WA 98195, USA; faustman@uw.edu; 10Agriculture and Economic Growth Team, Tetra Tech ARD, Burlington, VT 05401, USA; steveandjenp@gmail.com

**Keywords:** zoonotic disease, spillover, behavior change, community participatory approach, risk reduction interventions

## Abstract

**Background/Objectives**: The USAID STOP Spillover project in Cambodia aimed to reduce the risk of zoonotic virus spillover from bats to humans in bat guano farming communities. **Methods**: Using participatory tools, such as Outcome Mapping and Trials of Improved Practices, a team of local experts and community members collaboratively designed, tested, and refined biosafety and hygiene practices that are acceptable and sustainable to mitigate the risk of bat-borne disease spillover. We tracked progress and rolled out interventions to promote the adoption of safe behaviors that strengthen the understanding of zoonotic disease and reinforce the adoption of safety practices among bat guano producers and their neighbors. The intervention’s effectiveness was evaluated after three-month trials. **Results**: An improvement in knowledge, attitudes, and risk reduction practices was observed among participants. The primary motivators for adopting these measures were fear of disease, families’ well-being, cost savings, and experience of the COVID-19 pandemic. **Conclusions**: The community-driven approach fostered a sense of ownership, enabling participants to find the best solutions for their circumstance for long-term sustainability of the intervention. The findings recommended continued community engagement, improved access to biosafety and hygiene resources, and reinforced routine zoonotic disease surveillance. This model can be applied to mitigate emerging infectious disease spillover risks in similar contexts.

## 1. Introduction

Bats play crucial ecological roles in various ecosystems around the world, including pollination, seed dispersal, insect population control, and nutrient distribution, as bat guano is rich in nutrients and an excellent natural fertilizer [1,2]. In many parts of the world, including Cambodia, the latter feature has been exploited by human ingenuity through bat guano harvesting and utilization as agricultural fertilizer. While numerous benefits of guano use have been reported, bats are also known to harbor and transmit various zoonotic diseases, including coronaviruses and Nipah virus among others, that can pose significant risks to human and animal health [3,4,5,6,7]. The recent outbreaks of infectious diseases, including COVID-19, avian influenza, Nipah, and other diseases, in the Asia-Pacific region and globally, pose significant global economic and public health risks [8].

Bats can transmit infectious diseases to humans and other animals through various pathways, including direct contact, bat consumption, contaminated food or water, and exposure to contaminated environments [2,9]. High-risk behaviors that present these transmission pathways include bat guano harvesting, disturbing bat roosts, consuming fruits partially eaten by bats, drinking palm juice, hunting or consuming bats [7,10], and interacting with intermediate hosts like horses, pigs, civets, or non-human primates [9]. People who live adjacent to bat roosts and caves have frequent contact with bats and bat waste [11,12,13,14,15]. Women who work and live in close proximity to bats have been considered to be at greater risk since they have been found to be less knowledgeable about bats and the health risk they pose [16], while being more involved in bat guano harvesting activities and household work. In this regard, women may be a particularly vulnerable group for zoonotic disease transmission [17,18]. 

In Cambodia, the district of Kang Meas in Kampong Cham province has communities that have been working and living in close proximity with bats since before the Khmer Rouge era (1975–1979), when the practice of building bat roosts and collecting guano was documented [19]. Bat guano producers (BGPs) in these communities constructed artificial roosts in close proximity to their houses, livestock, and crops in order to attract insectivorous bats, guard the bats from any hunting, and facilitate the collection of bat guano to be sold as a natural fertilizer for crop and commercial farming activities [20]. Guano farming activities include constructing bat roosts, replacing palm leaves, installing plastic nets under the bat roosts, and guarding the bats in the roosts as well as collecting, selling, and using the bat guano. These activities put people and animals in close contact with bats, bat feces, and bat urine.

Underlying the emergence of diseases and the spread of epidemics is the behavior of individuals, the social structures in which they operate, and the political and economic environment that shapes outcomes. In order to prevent transmission, individuals need to change their current behaviors. Behavior change across all levels is fundamental to reducing spillover of zoonotic disease from wildlife such as bats. Many health behavior change models, like the Health Belief Model, incorporate risk perception as a key factor in predicting health behaviors. If someone perceives a high risk associated with a certain behavior, they are more likely to adopt preventive practices [21,22,23,24]. Several previous studies indicated that people who were aware of zoonotic viral infection displayed more concern about the risks of spillover and practiced better prevention measures such as washing hands with soap [25], wearing personal protective equipment [26], and vaccination [27].

Global health organizations have long focused on responding to outbreaks rather than preventing them. A community participatory approach is essential for preventing infectious diseases and promoting sustainable behavior change [28,29,30]. This approach promotes strong collaboration among public health, animal health, and environmental health personnel, as well as community stakeholders. Communities are at the center of public health emergencies. It is critical that the communities are actively engaged as partners to co-develop solutions, not only during outbreak responses, but also in preventing public health emergencies [31]. By raising awareness of zoonotic diseases and demonstrating self-protection measures, communities can gain a positive attitude and behavior of reducing the risk of infection from bats and other animals [15,26,32]. Additionally, community conversation can foster dialog among women and men, and stakeholders in communities, leading to the identification and prioritization of high-risk activities and solutions. Evidence-based results from these efforts can help identify feasible and acceptable solutions that contribute to positive changes in disease mitigation and prevention [33,34,35].

Despite the potential risks of bat-borne diseases, particularly coronaviruses, there is a significant knowledge gap regarding safety practices in rural Cambodian bat guano farming communities. Understanding factors influencing people’s decisions about virus prevention and the challenges hindering mitigation efforts is crucial. This knowledge will inform strategies that can reduce the risk of coronavirus transmission from bats to humans in Cambodia. The USAID STOP Spillover team in Cambodia used participatory tools, such as Outcome Mapping and Trials of Improved practices (TIPs), to engage local stakeholders from the planning stage to design, test, and implement interventions that are more sustainable to mitigate the risk of bat-borne disease spillover in bat guano farming communities in Cambodia [36,37]. This article provides insights into the feasibility and effectiveness of applying these participatory approaches in reducing the risk of zoonotic disease spillover in critical bat–human interfaces, as well as other high-risk interfaces.

## 2. Materials and Methods

Between 2022 and 2024, a four-phase implementation research project was implemented in bat guano farming communities to mitigate zoonotic disease risks (Figure 1). The planning and design phase used the Outcome Mapping methodology, a collaborative, stakeholder-driven approach, to engage a broad range of traditional and non-traditional stakeholders to identify and map desired outcomes [36]. Outcome mapping workshops, held first at the national level in Phnom Penh and then at the provincial level in Kampong Cham, engaged a range of local stakeholders to identify priority pathogens and high-risk interfaces for STOP Spillover to focus its initial efforts, as well as opportunities, gaps, and barriers. To address key knowledge gaps, a One Health working group was formed with the composition of nominated technical officers from pertinent government ministries at national and provincial levels who have authorization and technical expertise in zoonotic disease prevention and control. They conducted a formative mixed-methods study using a KAP survey, focus group discussions, key informant interviews, and direct observation among the bat guano producers and their neighbors in Kampong Cham to assess their current practices and the risks they face related to known risk pathways. This was followed by targeted sampling to assess the presence of infectious agents along these risk pathways (i.e., water, food, and household surfaces). The findings from those studies formed the basis for the Trials of Improved Practices (TIPs), a participatory formative research technique that pretests and refines practices or behaviors with a purposeful sample prior to launching them more widely to ensure that those practices are acceptable and feasible for those performing them [37,38,39]. TIPs has been used to finalize intervention designs to control malaria, other infectious diseases, and solid waste, and to improve the uptake of nutrition, family planning and water, sanitation, and hygiene (WASH) practices. TIPs focuses on behavior and what people can and are willing to do, in this case, to improve biosafety and hygiene practices to reduce the risk of spillover from bats to humans. The implementation phase strengthened the understanding of zoonotic risks among the bat guano farming communities and reinforced the adoption of improved biosafety and hygiene practices to reduce risk. The evaluation phase assessed behavior changes and the effectiveness of interventions in bat guano farming communities.

Women were included and represented in both the studies and interventions at national, provincial, and community levels. A stakeholder mapping exercise was conducted to identify relevant actors, including women working on bat guano farms. They were then invited to participate in the studies and dialogs and a series of interventions that enhance their behavior, practices, and safety.

### 2.1. Planning and Design Phase

#### 2.1.1. Outcome Mapping: National Stakeholder Engagement Meeting

In June 2022, following the steps of OM, a national stakeholder engagement meeting was conducted that brought together 64 stakeholders from pertinent government ministries, research and academic institutions, donor agencies, international experts, and NGOs (Step 1 of Figure 1). The stakeholders worked in groups and identified pathogens of concern: coronavirus and avian flu and then identified top three suspected high-risk interfaces: rodent-coronavirus in Kandal province, poultry-avian influenza in Phnom Penh, and bat-coronavirus in Kampong Cham province. To prioritize the importance of the zoonotic diseases, a scoring matrix was used, considering three key criteria: impact on human and animal health, ease of implementation of interventions, and stakeholder participation. According to the matrix, the bat-coronavirus scored highest and, based on the previous studies conducted by USAID PREDICT [20], Kang Meas district of Kampong Cham province was selected as the project implementation site. To ensure effective implementation, the involvement of local communities, including bat guano farmers, local authorities, and stakeholders in the subsequent stages of risk reduction activities was recommended and determined.

#### 2.1.2. Provincial Outcome Mapping

After identifying the priority interface “Bat-human” at the national stakeholders meeting, we conducted sub-national provincial meetings and site visits in July 2022. The engagement meetings established relationships with provincial, district, and community stakeholders and the observation of bat guano farmers during site visits showed that farmers often build artificial bat roosts near their homes, increasing their exposure to bats and guano. Additionally, they collect bat guano without wearing any personal protective equipment, raising the risk of disease transmission.

Following the meetings and site observation, we conducted a four-day provincial OM workshop in Kampong Cham province involving diverse stakeholders, including government officials, researchers, academics, NGOs, local authorities, health center staff, and bat guano farmers, especially women (Step 2 of Figure 1). By working in small groups and using matrix tables, the participants identified and prioritized opportunities, gaps, and barriers related to zoonotic disease and risk pathways in bat guano farming communities. The workshop resulted in the development of a shared vision for the community, identification of critical partners, outcome targets and risk reduction interventions, which aimed to improve biosafety and hygiene practices to prevent zoonotic disease transmission (Table 1).

#### 2.1.3. One Health Design, Research, and Mentorship Working Group Establishment

A One Health Design, Research, and Mentorship (OH-DReaM) working group was established in December 2022, bringing together the STOP Spillover consortium experts and in-country One Health specialists from the national and sub-national levels of the Ministry of Agriculture, Forestry, and Fisheries, Ministry of Health, and Ministry of Rural Development (Step 3 of Figure 1). To strengthen their roles and participation, eight working group members received training in zoonotic diseases, quantitative and qualitative research methodologies, and risk reduction strategies. The key activities undertaken by the working group included the following:Research and Knowledge Gap Assessment: Collaborative research to identify knowledge, attitude, and practice gaps among bat guano farmers and their neighbors.Intervention Design and Validation: Participatory designing, testing, and prioritizing feasible risk reduction interventions through discussions with community members.Implementation of Risk Reduction Interventions: Collaborative implementation of educational activities, household visits, and technical support to promote biosafety and hygiene practices.Monitoring and Evaluation: Joint monitoring and evaluation of intervention effectiveness and making necessary adjustments to strengthen risk reduction efforts.

### 2.2. Knowledge, Attitude and Practices Study

A study on knowledge, attitude, and practices (KAP) was conducted in March 2023, to assess potential risk pathways, and understand the behavior of BGPs and their neighboring non-bat guano producers (NBGPs) regarding zoonotic disease and spillover risk reduction and mitigation (Step 4 of Figure 1). This study utilized a mixed-methods approach, combining quantitative and qualitative data collection techniques. Kobo Toolbox was used for collecting household surveys, while an observation matrix was used for household observations. Focus group discussions (FGDs) and key informant interviews (KIIs) were guided by discussion guides and recorded by note-takers. Table 2 presents the number of respondents in each category.

The OH-DReaM working group members and community stakeholders, following rigorous training on research methodologies, participated in the research design and data collection and ensured the quality of the collected data and information. The analysis primarily relied on descriptive statistics. To account for data variability, 95% binomial confidence intervals were calculated for frequency data using R 4.1.3 (R Foundation for Statistical Computing, Vienna, Austria). Differences in proportions were analyzed using Fisher’s exact test [40]. Qualitative data was analyzed using an inductive approach to identify emerging themes from KIIs and FGDs. The data was coded by a primary analyst, with input and consensus-building from a team of qualitative researchers.

### 2.3. Sampling

In April and May 2023, two studies were conducted to detect coronaviruses in animal-derived and environmental samples. (Step 5 of Figure 1). The OH-DReaM working group members were trained in sampling techniques, biosafety, and biosecurity. In collaboration with STOP Spillover and local authorities, we collected 146 bat guano samples and 116 bat urine samples from 17 BGP households Appendix A. Environmental sampling teams collected 70 food samples, 75 water samples, and 376 surface samples from the household environments of 10 BGP and 10 NBGP households (see Appendix A).

Guano and urine samples were collected non-invasively by placing plastic sheets underneath the roosting sites. The fecal material captured on the plastic sheets was collected using plastic straws, while urine samples were obtained with sterile polyester swabs (Copan Diagnostics, Murrieta, CA, USA). In addition, food, water, and household surface samples were collected during the daytime from inside and outside the BGP and NBGP homes (Table 3). All collected samples were stored appropriately and securely transported on ice to the Institute Pasteur of Cambodia and the National Animal Health and Production Research Institute under the Ministry of Agriculture, Forestry, and Fisheries (MAFF) for laboratory analysis, following previously published protocols [41]. Briefly, guano samples were homogenized using a pellet pestle and pooled in groups of five (100 μL of suspension per sample). Suspension was concentrated using a 0.45 μL syringe filter (Thermo Scientific, Waltham, MA, USA). RNA was extracted using the Direct-Zol RNA MiniPrep kit (Zymo Research, Tustin, CA, USA) and cDNA transcribed using SuperScript III First-Strand Synthesis Super-Mix (Invitrogen, San Diego, CA, USA). Pan-Coronavirus conventional hemi-nested RT-PCR targeting the RdRp gene as performed as previously described [41]. Positive samples were sanger sequenced (Macrogen, Seoul, Republic of Korea) in both directions and sequences compared with available comparison sequences using the National Center for Biotechnology Information BLAST (https://blast.ncbi.nlm.nih.gov/Blast.cgi (accessed on 21 October 2024)) search. Positive pools were disaggregated and samples tested individually. Phylogenetic analyses were performed using Geneious Prime 2022.1.1 (Biomatters Ltd., Auckland, New Zealand). For more information, see Appendix A.

### 2.4. Intervention Testing Phase

#### Trials of Improved Practices

Using the TIPs method, bat guano farmers and their neighbors participated in trying and shaping improved practices to reduce bat–human interactions that are acceptable and feasible for them to do. TIPs were conducted over three weeks in August 2023, involving 13 BGP and 10 NBGP households (Step 5 of Figure 2). Three visits were made to each household using interview guides. The first visit was conducted to assess their current practices that put them at risk and to negotiate with each household to prioritize two to three new or modified behaviors to try to further reduce the risks they encounter in living and working with bats. Illustrated reminder sheets for each of the recommended practices were left with each household along with a calendar to use for tracking their progress during the trial period. The interviewer then returned midway for the second visit, using the reminder sheets to assess the progress, problem solve as needed, and encourage participants to continue. At the end, the interviewer visited each household and used final assessment tools to talk with them about their experience, assess the extent of change, and discuss the family’s reactions and intention to continue the practice (Figure 2).

The three-week trials involved 23 participating households to try all seven risk reduction behaviors. With disaggregation, 11 households chose wearing a full set of PPE and properly removing, cleaning, and storing them, 2 households tested safe storage of harvested guano, eight households practiced handwashing with soap and running water, especially after dealing with bats and guano, 11 households cleaned high-touch household surfaces daily with soap or disinfectant, 4 households dried and covered their food properly, and 9 households chose the implementation of safe dead bat disposal (Figure 3).

### 2.5. Implementation Phase

#### Intervention Roll-Out

Following the TIPs results, the STOP Spillover team, in collaboration with OH-DReaM Working Group and local helping hand (HH) group members, planned and implemented a series of education activities for a wider group of people. Figure 3 describes the design and implementation approaches carried out in bat guano farming communities.

Community Dialog: The OH-DReaM team and STOP Spillover staff conducted dialogs with community members to discuss and prioritize solutions to the challenges and concerns they faced over decades regarding the construction of artificial bat roosts next to their houses, causing bad smells and disturbances. The participants shared their ideas and exchanged good practices of implementing risk reduction measures at their farms and households. For instance, bat guano farmers brought the sets of the PPE they have used for guano collection to demonstrate to other participants, as an example, and discuss how comfortable they are.

Demonstration-based Education (DBE): DBE sessions were conducted by OH-DReaM and HH group members to strengthen the practical knowledge and skills, demonstrating the social and behavior change communication materials and answering clarifying questions. The DBE also guided participants on the techniques for wearing, removing, and storing the PPE, the steps for handwashing with soap, and how to use disinfecting materials.

Participatory monitoring and technical support: From the CD and DBE, the participants agreed and committed to continuing their biosafety and hygiene practices. The OH-DReaM and HH group members conducted monitoring visits at the households to observe progress, challenges, and gaps related to risk reduction practices. The team members discussed the results of monitoring visits with the community during the community dialog for further improvement and their commitment. The OH-DReaM and HH group members helped install handwashing stations, and guided them in identifying spaces for storing PPE and cleaning household yards for a few BGP households.

Peer-to-peer learning: The team facilitated exchange activities among BGP households to promote visual peer-to-peer learning and knowledge sharing. The exchange activities included the good practices of installing handwashing stations, storing PPE, and safe guano storage.

### 2.6. Evaluation Phase

#### Validation Assessment

The assessment was collaboratively conducted in June and July 2024 to evaluate the effectiveness and sustainability of the risk reduction interventions implemented in bat guano farming communities. A mixed-methods approach was used again to collect both quantitative and qualitative data (Table 4). The team exported data from Kobo Toolbox (version v.2024.1.3) to Excel, then cleaned and analyzed this data in Excel (v. 16.0) and Stata (v. 18.0), based on baseline data and validation assessment guidelines. For qualitative data from FGDs and KIIs, the team verified field notes with audio recordings. A thematic analysis was conducted to identify key themes aligning with quantitative findings.

The team also used an Adenosine Triphosphate (ATP) luminometer (Hygiena Systemsure) to test the level of contamination on household surfaces to measure the compliance with and efficacy of using water and soap to wipe down high-touch surfaces. This method employs a firefly luciferase reaction to quantify ATP, present in all living cells, as a measure of organic contamination. A representative sample was collected following the manufacturer’s instructions. Briefly, a predetermined 10 cm × 10 cm area surface inside the selected households was sampled with the included swab. The sample is inserted into the instrument for analysis and a reading, reported in Relative Light Units (RLU), is produced. After the first sample was collected, a second swab was taken on the same area after cleaning it with soap, in the manner promoted by the intervention, and letting it dry to compare results. In total, 60 samples were collected from 30 households both before and after.

### 2.7. Ethical Statement

The study protocols were designed and implemented in accordance with the ethics approvals both the Ministry of Health in Cambodia and Tufts University IRB in the United States. All participants were fully informed about the study’s objectives and assured of the confidentiality of their information and results. Prior to participation, all participants provided clear explanation and written informed consent and were made aware of their right to withdraw from the study at any time.

## 3. Results

### 3.1. Knowledge, Attitudes and Practices Study (KAP)

Table 5 provides demographic information on the KAP and validation assessment (its results are presented in later sections) with the same population. One respondent was selected from each household based on their level of involvement in bat guano farming activities. The KAP study included 67 representatives from 16 BGP and 51 NBGP households. The data reveals that older women (53–64 years), or equal to 62.5% of representatives, are mostly involved in bat guano farming activities. The majority of individuals have studied until the primary education level or below. Agriculture was the main source of income for most participants, while bat guano farming was the key source for one third of the BGP households. The majority of households reside less than 20 m away from all artificial bat roosts. In the 16 BGP households, a total of 10 children (7 boys and 3 girls) were reported to work in the bat guano activities.

The bat farming activities included collecting, screening, drying, packaging, storing, and selling the guano, installing plastic nets under the bat roosts to harvest the guano, constructing and reconstructing the bat roosts, and replacing the palm leaves. The community is accustomed to living near bats, does not consider them as hazards, and has never experienced diseases or risks of spillover effects from bats to humans (Table 6).


*“I don’t think it has an infection either. If it had been infected, I would have died quite a long time ago. But I think the government and foreigners are worried.” (IDI, M, bat guano farmer, 68 years old).*



*In fact, we have used Krama or caps for a long time when we enter the bat farm so that we can protect ourselves from becoming dirty with bat waste. (FGD, F, bat guano farmer)*



*“Before they didn’t use anything. Only a scarf or cap to cover the head. The cap is used to protect us from urine because of smell or drops of urine or guano from the bats, but I don’t think they protect us from disease or virus.” (IDI, M, CC, 67 years old)*



*“They [bat and non-bat guano producers] live together as usual, they understand each other. Some people complain about the noise and smell. I think they could not talk with bat guano producers directly, as they are afraid that those people will be angry. People murmur about this. I never received complaints officially from non-bat guano farmers and not all non-bat guano producers complain. For example, among 5 non-bat households, there might be 1 that complains. I think the rest are getting used to the smell.” (IDI, F, CCWC, 42 years old)*


### 3.2. Sampling

We detected alphacoronaviruses in bat urine and feces in 15 out of 17 bat guano farms. Additionally, alphacoronaviruses or infectious bronchitis virus (IBV) were found in food and on household surfaces in both BGP and NBGP households [42]. None of the water samples tested positive. The food sample testing positive was on coconut, and the surface samples testing positive were on bat roosts, outside tables, food cover, the kitchen table, and the upstairs table (Table 7).

### 3.3. Risk Reduction Intervention Design and Testing

#### Trials of Improved Practices

TIPs were co-designed and implemented by 23 households, choosing two to three new or modified behaviors that can best mitigate and reduce the risks they encounter. The seven risk reduction behaviors for the TIPs were informed by the KAP as well as the sampling. The results of the TIPs indicated that the majority of participants implemented the selected interventions and their sub-behaviors at close to optimal levels (Figure 4). As an example, 11 BGP households prioritized and implemented wearing a full set of PPE. This required them to properly practice all its sub-behaviors, including wearing hats or caps, eye protection, masks, gloves, long garments, and boots. The final assessment during the trial period indicated that all 11 BGP households implemented almost all of the sub-behaviors. For more information, see Appendix A.

Both BGPs and NBGPs adopted all the selected risk reduction behaviors. They were willing to invest their own resources in protective equipment and continue practicing biosafety and hygiene measures. The research team learned during the trials that some behaviors needed further refinement and improvement. For instance, some BGPs experienced discomfort with certain PPE items, such as boots and glasses. They had limited access to PPE due to the long distance required to reach markets, which was particularly difficult for older women. This inspired the idea of engaging local vendors to supply essential PPE locally in the community. Safe guano storage and dead bat disposal were the least adopted practices, suggesting limited health risk perception among some participating households. This gave the project team the ideas to strengthen the community’s awareness and reinforcement of risk reduction practices through innovative and collaborative approaches.


*“It’s like a reminder for me not to forget to protect myself”. (BGP_011)*



*“I will continue using them (PPE) to take care of my health.” (BGP_009)*



*“I don’t get much guano, so I mostly just collect and dry it and keep it in the open jar. I will put it in the common plastic bag if anyone buys that guano…sometimes the chickens from other HH jumped into that jar to eat insects, sleep and sometimes also lay eggs there…” (BGP_005)*



*“I will tell my daughters grandchildren to keep good hygiene by cleaning surfaces every morning with soap. We have to live in good hygiene because we are living close to the bat roosts.” (NBGP_011)*


### 3.4. Intervention Validation Assessment

#### 3.4.1. Behavior and Risk Reduction Practices

Table 8 presents data on key behaviors related to zoonotic disease prevention from the KAP study (before the intervention) and validation study (after the intervention). Before the intervention, awareness of zoonotic diseases was low, and only a small percentage of women made decisions in bat guano farm operations. Additionally, the use of protective equipment was limited, and safe practices like handwashing, food and water source covering, and safe bat guano storage were not widespread. Following the intervention, a trend toward improvement was observed in all key behaviors and a statistically significant improvement was documented in all but three key behaviors. Awareness of zoonotic diseases was high and women’s decision-making role in bat guano farming activities increased. The use of personal protective equipment increased significantly, while handwashing, food and water source covering, and safe bat guano storage practices were adopted by a large majority.


*“Handwashing is the most important because after we work, we need to wash. If we don’t wash, we touch food and eat so it will affect us. In short, whenever we finish the guano collection, we need to shower and wash our hands immediately.” (FGD_BGP_Male)*



*“Raising the net is good as it is easier than before. It also protects from chickens and dogs as the dog really likes playing in the bat guano and protects them from carrying out the infectious diseases. It also saves time in collecting the guano as well.” (FGD_BGP_Female)*



*“Last time the bat flew into my water container, so in the morning I took them and buried them. I also throw away that water without giving it to my cow.” (FGD_NBGP)*


#### 3.4.2. Risk Reduction Effectiveness

Both BGP and NBGP households expressed strong belief that the risk reduction measures they have practiced helped them and their communities stay safer from disease spillover from the bats. As an example, the majority of respondents indicated that safe guano storage reduced direct contact with the guano and prevented animals from contacting the bat roosts, reducing the disease spillover risk from those animals. In the focus group discussion, they expressed that the use of PPE could prevent them from infectious disease, but adding body and hand hygiene would fully help to protect them.

Table 9 presents the level of cleanness on various household surfaces between before and after wiping with soap, based on the ATP bioluminescence analysis. On average, the level of soiling decreased 79.3% after cleaning. The reduction level varied across different surface types, with brick meal tables showing the highest reduction (91.8%) and wooden meal tables exhibiting the lowest (56.1%).


*“The use of full PPE may protect about 90% but not 100% as we need to shower with soap. If the disease is serious, it can also protect some, but it is not a serious disease it can protect fully.” (FGD_BGP_Female)*



*“The project educated us on zoonotic diseases and prevention techniques. We understood and applied it, so we feel confident about 90% that we are safer from the spillover risk. We are confident enough to be working at the farms.” (FGD_BGP_Male)*


### 3.5. Behavior Change Motivators

Both BGP and NBGP households were more open to changing their biosafety and hygiene practices after joining in a series of risk reduction activities. Figure 5 shows that 55 respondents were concerned about their own health, families’ well-being, spreading the disease, and incurring cost and loss of income. Similar views were expressed in focus group discussions and the impact of the COVID-19 pandemic was mentioned in this context. The discussions revealed some differences in the priority for their concerns. The women in the group mentioned families’ well-being as a motivator while the men expressed concerns over personal health and cost.


*“We are changing because the project tells us. Then why do we change? Because we are afraid of infectious diseases. Especially after we see COVID. We are afraid there will be some diseases like COVID to happen again.” (FGD_BGP_Female)*



*“COVID is also an example of the disease, and they learn and react to this quickly.” (IDI_PDA)*



*“We are afraid of viruses from bats for our health and if it affects us and will affect our family and other people in this community.” (FGD_BGP_Female)*



*“We change because we take care of our health. [Being] sick is never easy. Being sick also costs so much money. Whenever we are sick, it will be gone even with how much money we have (spending on health care). When we are sick we will lose our jobs and money. So just to protect our health is easier…” (FGD_BGP_Male)*


### 3.6. Risk Reduction Sustainability

The bat guano farmers and their neighbors expressed satisfaction with the interventions, committed to continuing the risk reduction practices and committed to sharing these good practices with others even after the project ended. Figure 6 indicates the different risk reduction practices implemented that respondents were committed to continuing. In the FDGs and KII, BGP, provincial stakeholders, and local administrations showed willingness to practice and extend awareness education and technical support to the people by integrating these initiatives into their community investment and development planning and implementation.


*“We will continue practicing hygiene even when the project phases out because the project taught us. We have knowledge and we will practice because we want to prevent infectious diseases and protect our health. We will be healthy if we can avoid disease and we also can save more money.” (FGD_BGP_Female)*



*“I plan to make a report and report the results to the commune and local authority to ask them to take over and take care these activities…The most important thing is the support of local authority…For me, I am still committed to come over this area to see how the activities are and continue to raise awareness among them or remind them…I think if we can bring the result to provincial level so that we can ask provincial level to take over…” (IDI_provincial government representative)*



*“Monks and priests here have knowledge; I also have knowledge so I will continue to share this knowledge and information. Even if I have a new monk coming, I will share with them. I will continue mainstreaming and reminding people through events in the pagoda every full moon day and ceremonies in the village.” (IDI_community representative).*


## 4. Discussion

### 4.1. Community Participatory Approach

USAID STOP Spillover’s community-level-risk reduction interventions significantly enhanced the understanding of zoonotic disease and strengthened biosafety and hygiene practices among people living in bat guano farming communities in Cambodia. Integral to this achievement was the community participatory approach, which actively involved the local community right from the project formulation stage and throughout implementation, an approach that aligns with previous studies concluding that the sustainability of donor-funded projects is strongly influenced by community participation [43,44]. In contrast to many donor-funded development projects in which experts disappear and the projects collapse once the donor withdraws, the members of the OH-DReaM working group, which include local government agencies and local authorities, appear likely to continue the work, having honed their skills and taken ownership of the intervention. The participation of the local Helping Hands group, which included the local authorities, Buddhist monks, pagoda committee members, and a local vendor, also contributed to supporting the adoption of the risk reduction practices among the community members, especially men and women among bat guano farmers.

Through the Outcome Mapping planning sessions, national and sub-national stakeholders prioritized the bat–human interface in bat guano farming communities in Kampong Cham province as an important area of bat-borne disease spillover risk. Local stakeholders, including bat guano farmers, were engaged in all stages: from identifying risks to designing, testing, and implementing interventions. This participatory approach ensures that interventions are relevant, feasible, and sustainable and align with earlier work on the ecological benefits of bats and the risks they pose [1,6].

### 4.2. Understanding Current Practices and Risk Pathways

Understanding the knowledge gaps, current practices, and risk factors of the bat guano farmers and their community was instrumental to designing an effective intervention. Building on the previous work of USAID PREDICT, the KAP study identified the most important risk pathways for zoonotic spillover in bat guano farming. These pathways included the close proximity of bat roosts to homes, frequent human–bat–livestock interactions, and potential contamination of food and water. Limited knowledge regarding bat-borne diseases, PPE use, and risk reduction behaviors further exacerbated spillover risks [11,12,13,14,15,45]. Notably, women often performed important tasks in guano production yet held minimal decision-making power about implementing safety measures or how to spend resources, highlighting the need for gender-sensitive interventions.

Ensuring a participatory approach that simultaneously addresses the tradition of generating livelihoods through bat farming and health risks is vital for the sustained adoption of safer practices [26,32]. Social and behavior change (SBC) strategies, tailored to distinct gender roles in bat farming, can significantly improve long-term protective behaviors.

The focus on identifying the potential zoonotic bat-borne disease risk associated with coronaviruses in bat guano farms and household environment through laboratory testing further helped to design targeted interventions. Findings revealed a high prevalence of *Alphacoronaviruses* in bat urine and feces across multiple farms, underscoring zoonotic spillover potential. Virus detection on household surfaces—including food covers, tables, and coconuts—suggested the possibility of livestock acting as mechanical vectors. These results closely matched the findings regarding the same group of viruses in the same species of bats in a farm 200 km south in Dong Thap province, Viet Nam [42], emphasizing the role of environmental contamination in enabling indirect human exposure.

While water sources tested negative, the presence of virus was found on everyday objects, inside and outside the houses, especially on the kitchen table and other high-touch surfaces. This raises concerns about the potential for direct or indirect transmission of the virus to humans and calls for continued monitoring, surveillance, and detailed phylogenetic analyses to understand viral transmission dynamics.

### 4.3. Trials of Improved Practices (TIPs)

TIPs built upon KAP findings and biological sampling results by encouraging households to select and adopt two or three feasible risk reduction behaviors. This intervention demonstrated high levels of adoption for selected risk reduction behaviors among participating households, often reaching near-optimal levels. For instance, a substantial number of households successfully implemented the use of full sets of PPE. Both BGPs and NBGPs expressed a willingness to invest in PPE and continue practicing improved biosafety and hygiene measures. Community engagement and participatory fact-finding effectively addressed mistrust towards external researchers and government officials, laying the groundwork for positive behavior change. This approach mirrors the success of similar initiatives in Southern Cameroon [46] and Myanmar [47], where community involvement was crucial in mitigating zoonotic spillover risks. However, there were challenges, such as discomfort with certain PPE items and lower uptake of safe guano storage or dead bat disposal. Addressing the affordability, practicality, and comfort of these practices were crucial for long-term adoption.

### 4.4. Validation Assessment

Formative studies shaped the community-driven, collaborative approach of promoting the adoption of safe and protective behavior and was key in designing a successful intervention that focused on accessibility, feasibility, affordability, and sustainability. The intervention validation assessment demonstrated the effectiveness of STOP Spillover in improving knowledge, attitudes, and risk reduction practices in bat guano farming communities. Notably, women, who are deeply involved in bat guano farming activities, emerged as proactive in adopting protective measures. Figure 7 highlights a three-stage behavioral change process: knowledge gain, increased cautiousness about disease transmission, and the adoption of new risk-minimizing behaviors. STOP Spillover strategic activities, such as community dialogs, demonstrations, household monitoring, technical support, and peer-to-peer learning, effectively conveyed knowledge and reinforced the adoption of recommended preventive practices.

Key motivators for behavior change shifted from knowledge improvement to a strong concern for personal and family health, driven by fears similar to those experienced during the COVID-19 pandemic. The community’s experience with the pandemic heightened their awareness of zoonotic diseases, reinforcing the importance of biosafety measures. This is consistent with the increased cautiousness observed in response to other zoonotic disease outbreaks, such as Nipah virus in Malaysia [43] and Ebola in West Africa [44]. The study also demonstrated the high effectiveness and sustainability of the implemented interventions. Tailored to the community’s specific needs and practices, the interventions, such as community dialogs and practical demonstrations, were easily understood and implemented. These participatory approaches were well-received and widely adopted, leading to significant improvements in practices like hand hygiene, PPE use, and safe guano storage.

Despite the positive outcomes, several challenges remain in ensuring the long-term effectiveness and sustainability of spillover risk reduction efforts. These include maintaining consistent community education, ensuring the accessibility and affordability of PPE and hygiene materials, and strengthening monitoring and support systems [17]. To address these challenges, continued and expanded educational activities are crucial, with a focus on empowering local communities to lead these efforts. This includes promoting knowledge sharing among community members, emphasizing the long-term benefits of adopting the best practices, and providing clear information on accessing healthcare resources. Furthermore, a robust surveillance system is essential for early detection and prevention of disease outbreaks. This requires active participation from all stakeholders, including local communities, and a strong focus on data collection, capacity building, and inter-institutional coordination.

## 5. Conclusions

In this article, we presented the entire life cycle of the USAID STOP Spillover project in Kampong Cham Province which demonstrated that community-led, participatory approaches can be highly effective in reducing zoonotic spillover risks in Cambodian bat guano farming communities. A key strength of the project was the strong focus on community participation, engaging local stakeholders throughout all phases of the project. This participatory approach, aligned with the One Health framework, fostered interdisciplinary collaboration and ensured that interventions were relevant, feasible, and acceptable to the communities, making the prospect of sustainability more likely. By prioritizing local engagement and collaboration through the OH-DReaM working group, the project ensured that interventions were both practical and culturally attuned. The KAP and biological studies highlighted the importance of hygiene and biosafety behaviors, especially given the high prevalence of alphacoronaviruses in bat guano farms. The TIPs methodology was instrumental in translating these insights into relevant risk reduction strategies, with the community selecting and refining feasible actions—from consistent PPE use to safer guano storage. Evaluations indicated that fear of disease, along with recent memories of COVID-19 experiences and women’s key role in bat guano production, were key drivers of behavior change.

To ensure the long-term sustainability of these efforts, the project recommends maintaining consistent community education, ensuring the accessibility and affordability of PPE, strengthening monitoring and support systems, and fostering community-led initiatives for knowledge sharing and behavior change. A robust surveillance system, incorporating active community participation, is essential for the early detection and prevention of future disease outbreaks. Future efforts to reduce the risk of zoonotic spillover from animals to humans at high-risk interfaces should consider meaningful community engagement and participatory approaches from project initiation and throughout.

## Figures and Tables

**Figure 1 tropicalmed-11-00007-f001:**
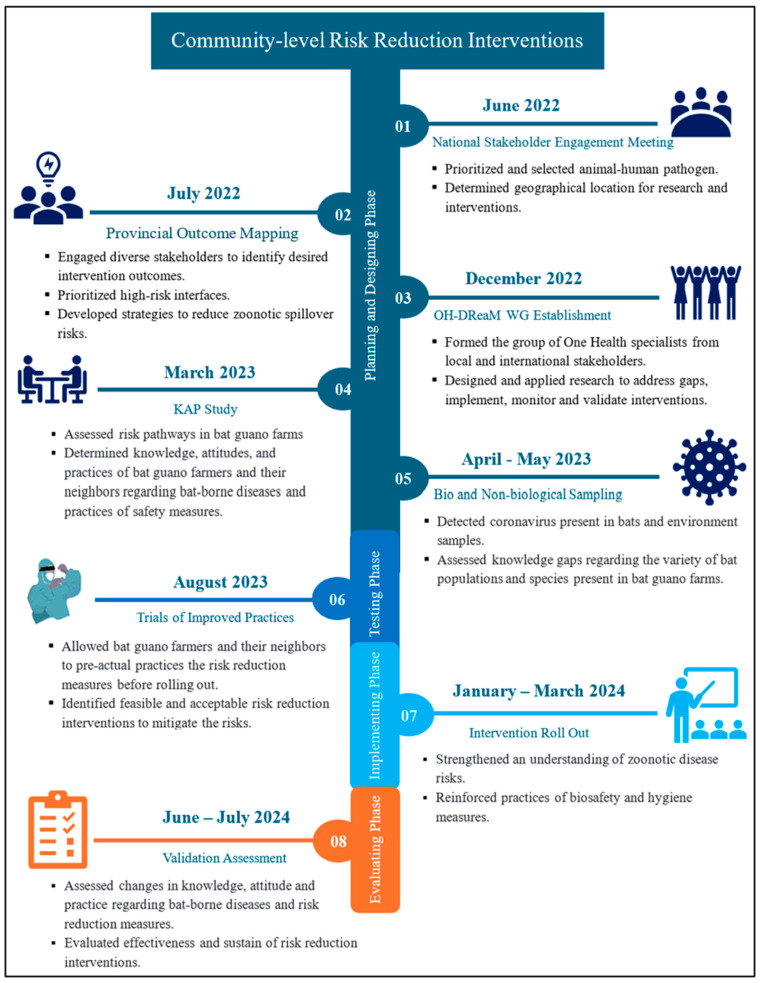
Phases of Community-level Risk Reduction Interventions.

**Figure 2 tropicalmed-11-00007-f002:**
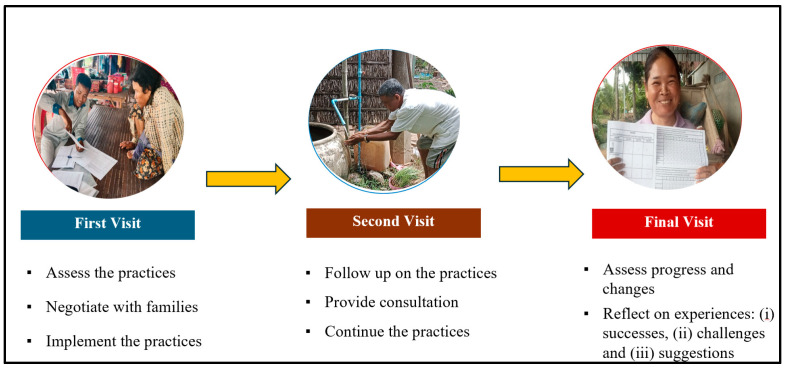
Three visits to the participating households during the trials period.

**Figure 3 tropicalmed-11-00007-f003:**
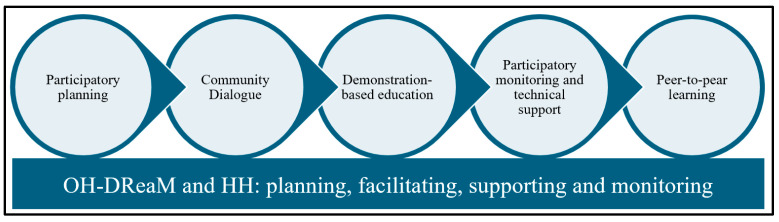
Participatory risk reduction intervention approach in bat guano farming communities.

**Figure 4 tropicalmed-11-00007-f004:**
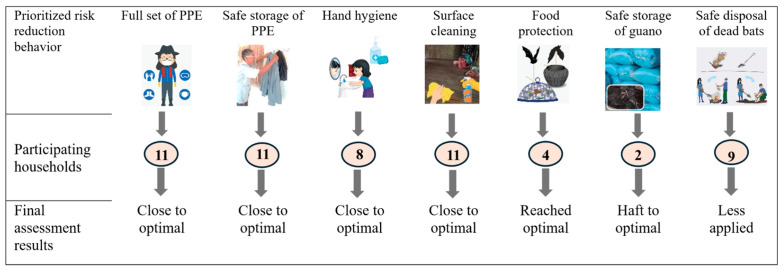
Seven risk reduction behaviors implemented by 23 households during trials period. Numbers in the circles are the number of households electing to participate in that practice.

**Figure 5 tropicalmed-11-00007-f005:**
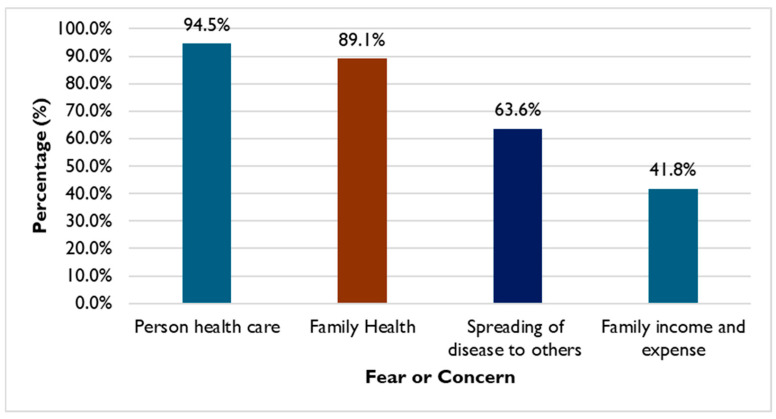
Behavior change motivators among 55 household respondents.

**Figure 6 tropicalmed-11-00007-f006:**
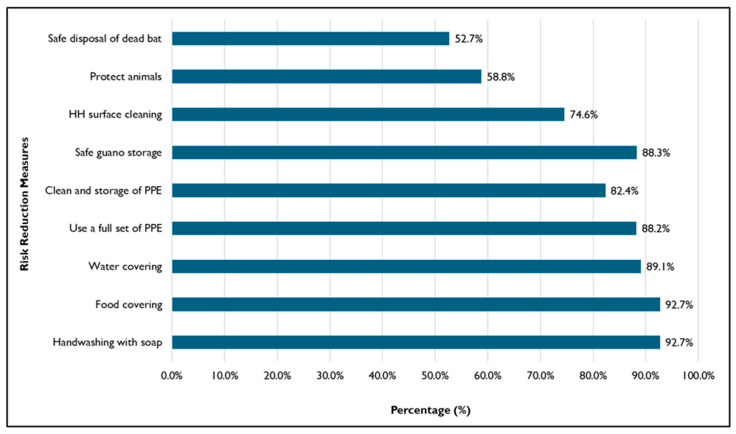
Percentages of 55 respondents committing to continuing the practice.

**Figure 7 tropicalmed-11-00007-f007:**
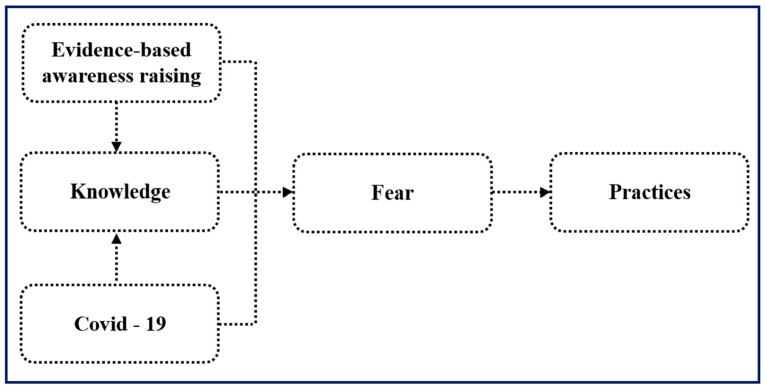
Pathway to behavior change among people in bat guano farming communities.

**Table 1 tropicalmed-11-00007-t001:** Four priority strategies suggested by stakeholders during the OM workshop.

(1)	Conduct research on bat ecology and pathogens prevalence to address knowledge gaps in bat guano farming communities;
(2)	Conduct a comprehensive national risk assessment to identify and map high-risk bat–human interfaces beyond guano harvesting;
(3)	Implement community-level risk reduction interventions to improve biosafety and hygiene practices among bat guano farmers and communities; and
(4)	Facilitate the coordination and capacity building of sentinel surveillance teams to monitor for spillover of coronaviruses and other bat-transmitted pathogens.

**Table 2 tropicalmed-11-00007-t002:** Data Collection Approach, Respondents, Sample Sizes, and Focus Applied in the KAP study.

Approach	Respondents (Type and Number)	Focus
Household Survey (Structured questionnaire)	67 households – 16 BGPs – 51 NBGPs	– Knowledge, attitude, and practices related to zoonotic diseases and risk reduction measures. – Risk pathways in bat guano farms and household livelihood activities. – Human–bat–livestock interactions.
Observations (Checklist)	15 households – 9 BGP households – 6 NBGP households	– Biosafety and hygiene practices in bat guano farms and household livelihood activities. – Human–bat–livestock interactions.
FGDs (Semi-structured questions)	15 participants – A BGP men group (5 persons) – A BGP women group (6 persons) – An NBGP men and women (mixed) group (4 persons)	– Risk factors and implications with human behavior by group and gender. – Concerns, suggestions and commitment related to zoonotic diseases, interventions and risk reduction practices.
KIIs (Semi-structured questions)	5 representatives – A BGP men representative – A BGP women representative – A Health Center head – A Commune Council – A commune committee for women and children	– Risk factors and implications with human behavior by group and gender – Concerns, suggestions and commitment

**Table 3 tropicalmed-11-00007-t003:** Environmental samples collected from BGP and NBGP households.

Type of Samples (n)	Prioritized Samples Collected
Food (70) * BGPH: 30 † NBGPs: 40	Dried banana, coconut waste, dried fish, dried pork, green vegetables (leftovers), jackfruits, leftover fish, leftover pork, mango, orange, potato, raw meat, rice, sugar cane, tomato, and vegetable waste/garbage.
Household surface (376) BGPHs: 208 NBGPs: 168	Bat roosts, basket over food, ceramic container for bat guano, clothes (near bat roost, in and out house), cooking table, cover on rice, hat for collecting bat guano, outside table, plate (inside in kitchen uncovered and outside kitchen), railing, table, fridge, table near stove, toilet door, upstairs floor, upstairs table, and water containers surfaces (outside and in the kitchen).
Water (75) BGPHs: 40 NBGPs: 35	Drinking water (39samples), pond water used for vegetable gardening (36 samples)

* BGPH: Bat guano producing households, † NBGPs: Non-bat guano producing households.

**Table 4 tropicalmed-11-00007-t004:** Research focus and sample sizes (n = 55).

Approach	Respondents (Type and Number)	Focus
Household Survey (Structured questionnaire)	55 households – 17 BGP – 38 NBGP	– Knowledge, attitude, and practices related to zoonotic diseases – Human–bat–animal interactions – Effectiveness and sustainability of risk reduction practices
Observations (Checklist)	55 households – 17 BGP – 38 NBGP	– Biosafety and hygiene practices in bat guano farms and household livelihood activities – Human–bat–animal interactions
FGDs (Semi-structured questions)	20 participants – A BGP men group (6 persons) – A BGP women group (5 persons) – A NBGP men and women group (9 persons)	– Knowledge, perception and practices – Effectiveness and commitment to sustain risk reduction practices – Effectiveness and commitment to sustain risk reduction practices
KIIs (Semi-structured questions)	6 respondents – A staff of the Provincial Department of Agriculture, Forestry and Fisheries (PDA) – A Health Center head – A Commune Council – A Buddhist monk – A Pagoda committee member – A local vendor	– Knowledge, perception and practices – Factors influenced the behavior change – Effectiveness and commitment to sustain risk reduction practices
Household surface sample testing	60 samples – 30 surface samples in 15 BGP households – 30 surface samples in 15 NBGP households	– Table: on break, meal, metal and wooden table – Water jar covers (Zinc)

**Table 5 tropicalmed-11-00007-t005:** Demographic data of the KAP study and validation assessment.

	KAP (n = 67)		Validation (n = 55) Like the New Demographic Tables	
	BGP (n = 16)	NBGP (n = 51)	BGP (n = 17)	NBGP (n = 38)
Sex				
– Male	7	23	7	8
– Female	9	28	10	30
Average Age (years)				
– Male	62	53	63	64
– Female	58	56	56	60
Education				
– None and Primary	16	34	12	28
– Secondary	0	17	5	10
Main income source				
– Agriculture	10	24	8	16
– Bat guano farm	5	0	7	0
– Laborers/workers	0	18	0	13
– Government/Business	1	9	2	8
Distance home to bat roost				
– 1–20 m	16	21	17	24
– 21–40 m	0	14	0	14
– 41 m up	0	16	0	0
Workforce at bat farm (total number)				
– Men	28	0	12	0
– Women	23	0	13	0
– Boys	7	0	2	0
– Girls	3	0	1	0

**Table 6 tropicalmed-11-00007-t006:** Behavior of BGPs (n = 16) and NBGPs (n = 51) regarding biosafety and hygiene practices.

Direct Exposure (n = 16)	Description of Results
Use of PPE (n = 16)	– 10.4% (7) of BGPs and NBGPs consumed bats. – 93.4% (15) of BGPs did not practice safe storage of harvested guano. – 87.5% (14) of BGPs did not practice safe disposal of dead bats (burning or burying).
Hygiene (n = 67)	– 75% (12) of BGPs did not wear full PPE when dealing with bats or guano. – 68.7% (11) of BGPs occasionally washed PPE with soap between uses, but sometimes they might just dry it or throw it away. – 43.7% (7) of BGPs changed clothes after working with guano.
Contamination (n = 67)	– 81.2% (13) of 16 BGPs washed hands with soap after handling guano. – 87.5% (14) regularly took a bath with soap immediately after any activity at the bat guano farm.
Human–bat–animal interaction (n = 16)	– 19.4% (13) covered food from contact of bats and livestock. – 44.8% (30) of households did not cover water consumption containers.
Bat–animal interaction (n = 16)	– 18.7% (13) did not practice techniques to prevent animals from roaming under bat roosts.
Gender (n = 16)	– 43.7% (7) of women led decision-making on bat guano farming operation.

**Table 7 tropicalmed-11-00007-t007:** Coronavirus PCR testing results.

No	Types of Samples	No. of Sample	Lab Results (%)	Detected Virus
**1**	Bat feces	146	18%	Alphacoronavirus
**2**	Bat urine	116	16.3%	Alphacoronavirus
**3**	Food	70	1.4%	Alphacoronavirus/IBV
**4**	Household surface	376	2.9%	Alphacoronavirus/IBV
**5**	Water	75	0	

**Table 8 tropicalmed-11-00007-t008:** Improvement of knowledge and practices before and after interventions.

Key Behaviors	Before Intervention (KAP)	*p*-Value	After Intervention (Validation)
BGPs’ and NBGPs’ awareness of zoonotic diseases	19.4% (13/67)	**<0.001**	100% (51/51)
BGP women-led decision-making on bat guano farm operations	43.7% (7/16)	0.732	52.9% (9/17)
BGPs wore a full set of protective equipment	25% (4/16)	**0.002**	83.2% (14/17)
BGPs’ handwashing with soap after contact with bats, guanos and urine	81.2% (13/16)	1.000	88.2% (45/55)
BGPs’ and NBGPs’ covering of foods from contact of bats and livestock	19.4% (13/67)	**<0.001**	98.2% (54/55)
BGPs’ and NBGPs’ covering of water sources from contacts bats and livestock	44.8% (30/67)	0.070	62.5% (34/55)
BGPs’ safe storage of harvested bat guano	6.6% (1/16)	**0.017**	47.1% (8/17)
BGPs’ and NGBPs’ wiping of household surfaces with soaps or disinfectant	9% (6/67)	**<0.001**	85.5% (47/55)
BGPs’ and NBGPs’ safe disposal of dead bats (burning or burying)	12.5% (2/16)	**<0.001**	75% (15/20)
BGPs’ prevention of domestic animals from roaming under the bat roosts	18.8% (3/16)	**<0.001**	82.4% (14/17 *)

* The sample size (n) for each indicator differs. We have presented numerators and denominators. **Bold** signifies results that were statistically significant at *p* < 0.05.

**Table 9 tropicalmed-11-00007-t009:** Results of household surfaces sampling before and after wiping with soap.

Type of HH Surfaces	No. of Samples	Level of Soiling (Average)		Reduction (%)
		Before	After	
1. Brick meal tables	16	2901	237	91.8%
2. Metal kitchen cabinets	2	1521	162	89.4%
3. Metal meal tables	6	1205	139	88.4%
4. Water jar covers (Zinc)	4	3197	855	73.1%
5. Wooden relaxing tables	26	2096	742	64.6%
6. Wooden meal tables	6	561	246	56.1%
Mean	60	11,481	2381	79.3%

## Data Availability

This research has various supporting data sources that were carried out in bat guano farming communities. These include the analyzed data and archived dataset, which are available upon the request and via the link below: https://drive.google.com/drive/folders/1XEKqBgkQ6FBYqWJSUIByJEhrsfnt5qGe.

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
