# Peer review of "Community Participatory Approach to Design, Test, and Implement Interventions That Reduce Risk of Bat-Borne Disease Spillover: A Case Study from Cambodia"

_tropicalmed, 2025, doi:10.3390/tropicalmed11010007_

Round 1

Reviewer 1 Report

Comments and Suggestions for Authors

Sok et al present an comprehensive community study aiming at reducing the overall risk of zoonotic pathogen spillover from bats in Cambodia. Congratulations on this important work! I have a few points that would need to be addressed. 

line 212 ff, table 7 and other occurrences, the samples should not be divided in biological and non-biological, these are all biological samples, suggest to change throughout to animal vs environmental or similar

line 234, pathogen detection methods are only described as 'following standard protocol'. This complete lack of detail is not acceptable. Please describe in Methods or refer to previous publication.

Table 6, 'Gender n=16' appears to be duplicated

Author Response

Comment 1: Sok et al present an comprehensive community study aiming at reducing the overall risk of zoonotic pathogen spillover from bats in Cambodia. Congratulations on this important work! I have a few points that would need to be addressed. 

Response 1: Thank you

Comment 2: line 212 ff, table 7 and other occurrences, the samples should not be divided in biological and non-biological, these are all biological samples, suggest to change throughout to animal vs environmental or similar.

Response 2: The use of this terminology has been revised throughout the manuscript. The two sample types are now generally described as ‘animal-derived’ and ‘environmental’. When combined, these two sample types may be referred to as ‘biological’.

Comment 3: line 234, pathogen detection methods are only described as 'following standard protocol'. This complete lack of detail is not acceptable. Please describe in Methods or refer to previous publication.

Response 3: Further description of the laboratory methods has been added (lines 245-256).

Comment 4: Table 6, 'Gender n=16' appears to be duplicated

Response 4: It is revised from Gender to “bat - animal interaction”. Please refer to line 386.

Reviewer 2 Report

Comments and Suggestions for Authors

This is a well written research article comprised of sound science.  I really have no edits or comments except for:

  1. Line 135 - I know that WASH is listed in the list of abbreviations but it might be helpful to the reader to spell it out when it is first introduced in the text.  You do this for all other abbreviations.

Nice paper!

Author Response

Comment 1: Line 135 - I know that WASH is listed in the list of abbreviations but it might be helpful to the reader to spell it out when it is first introduced in the text.  You do this for all other abbreviations.

Response 1: Water, sanitation and hygiene (WASH) was added. Please refer to line 137. 

Reviewer 3 Report

Comments and Suggestions for Authors

This assessment is a broad presentation of many components of public health interventions and educational initiatives to address individual human behaviors that may facilitate zoonotic spillover from bats to humans. Overall, this paper is very well written, the analyses and methods are robust, and the results are incredibly important for the broader One Health/international development community. My comments are minimal and are mostly focused on a few details that I think would benefit from further explanation and/or discussion. My largest questions centers upon how you ensured that women were well represented in your study participants/stakeholders (i.e. what did stakeholder identification look like for this project), since there was a special emphasis on women as an at-risk group.

Specific comments are as follows:

  • Introduction lines 61-64- you make this note about women being at greater risk, and follow it up in the discussion section, which makes sense. Nevertheless, I would suggest including a section either in the introduction about how gender was incorporated into the collaborative study design process, or add a short section into the methods about how you insured that women were well represented in your study participants and stakeholders. I would also recommend tying this in to your comment in the discussion section (lines 525-527).
  • Materials and Methods line 121: you indicate a range of local stakeholders that identified pathogens of high-risk. Are these the same stakeholders listed in 146-149? What was the composition of these stakeholders if not? My only question is if this group of stakeholders had any public health officials or local experts that could have aided in the conversation, or if risk perception of stakeholders may have shaped the pathogens that were considered.
  • Line 123-127: Who was represented in this working group?
  • Lines 211-214: How were the confidence intervals applied? In R or Excel? What software did the analyst use for coding?
  • Line 303: What version of Kobo Toolbox was used?
  • Results line 331-332: By “mostly” what percentage of respondents was this? Were there statically more women in bat guano activities?
  • Line 334-335: Less than 20 meters away from artificial roost, natural roosts, or both?
  • Line 411-413: Use of the word “significantly” here may imply a statistical test, which I don’t think you did unless I’m mistaken. Perhaps a statistical comparison is warranted?
  • Lines 434-437: I think more detail may be needed on the methods used for the bioluminescence analysis. How is level or soiling scaled? What is the upper bound here and what are the metrics? I see the reduction in Table 9, but was the before level or soiling particularly high? What types of pathogens appear in this type of analysis? This is not a method of assessment I am familiar with, and I worry that other readers may be unfamiliar as well.
  • Discussion lines 542-545: I would recommend specifically bringing up the kitchen table here. That result made me audibly gasp, and it really highlights the spillover risk here.

Author Response

Comment 1: Introduction lines 61-64- you make this note about women being at greater risk, and follow it up in the discussion section, which makes sense. Nevertheless, I would suggest including a section either in the introduction about how gender was incorporated into the collaborative study design process, or add a short section into the methods about how you insured that women were well represented in your study participants and stakeholders. I would also recommend tying this in to your comment in the discussion section (lines 525-527).

Response 1: The manuscript has been updated in the Method section on line 144 - 149, 181 and in the Discussion section on line 539. 

Comment 2: Materials and Methods line 121: you indicate a range of local stakeholders that identified pathogens of high-risk. Are these the same stakeholders listed in 146-149? What was the composition of these stakeholders if not? My only question is if this group of stakeholders had any public health officials or local experts that could have aided in the conversation, or if risk perception of stakeholders may have shaped the pathogens that were considered.

Response 2: The representatives of national stakeholders from the national level who joined the national and provincial OM workshop were the same. Additionally, the provincial stakeholders who joined the provincial OM workshop were new and different (they did not join the national OM). They are the technical staff from relevant government ministries, researchers from academic and research institutions, and international experts who have expertise in zoonotic diseases, bats and work relatively in this field.

Comment 3: Line 123-127: Who was represented in this working group?

Response 3: See the updates on the line 123 - 126. They  are the nominated technical officers from pertinent government ministries at national and provincial levels who have authorization and technical expertise in zoonotic disease prevention and control. 

Comment 4: Lines 211-214: How were the confidence intervals applied? In R or Excel? What software did the analyst use for coding?

Response 4: We have added this information to the methods section (lines 222-224). 

Comment 5: Line 303: What version of Kobo Toolbox was used?

Response 5: it is version v2024.1.3. See in line 326.

Comment 6: Results line 331-332: By “mostly” what percentage of respondents was this? Were there statically more women in bat guano activities?

Response 6: It is 62.5% that older women work more at bat guano farm in terms of frequency and longer period. See the update in line 358.

Comment 7: Line 334-335: Less than 20 meters away from artificial roost, natural roosts, or both?

Response 7: Only artificial bat roosts are present. See the update in line 362.

Comment 8: Line 411-413: Use of the word “significantly” here may imply a statistical test, which I don’t think you did unless I’m mistaken. Perhaps a statistical comparison is warranted?

Response 8: Fisher’s exact test results have been included for the uptake of key behaviors. References to ‘significant’ differences have been revised to ensure clarity and alignment with the results. 

Comment 9: Lines 434-437: I think more detail may be needed on the methods used for the bioluminescence analysis. How is level or soiling scaled? What is the upper bound here and what are the metrics? I see the reduction in Table 9, but was the before level or soiling particularly high? What types of pathogens appear in this type of analysis? This is not a method of assessment I am familiar with, and I worry that other readers may be unfamiliar as well.

Response 9: The statement in the Method section has been updated. This method is widely used in food safety and industrial hygiene. It was used in this case to assess the level of organic contamination on those household surfaces, not to detect a specific pathogen. We argue that cleaning household surfaces regularly and correctly with soap and or disinfectant could highly reduce this contamination and with it, the transmission risk of viruses. We have attempted to clarify the rationale for the use of this method in the text. See the updates in line 330 & 331.

Comment 10: Discussion lines 542-545: I would recommend specifically bringing up the kitchen table here. That result made me audibly gasp, and it really highlights the spillover risk here.

Response 10: Agreed. This section has been updated specifically with the kitchen table and other high-touch surfaces. See the updates in line 574.